# RESERVE OUTPUT UNITS FOR DEEP OPEN-SET LEARNING

**Tegan Maharaj**
MILA (Montreal Institute for Learning Algorithms)
Ecole Polytechnique de Montreal
`tegan.maharaj@polymtl.ca`

**David Krueger**
MILA (Montreal Institute for Learning Algorithms)
University of Montreal
`david.krueger@umontreal.ca`

## ABSTRACT

Open-set learning poses a classification problem where the set of class labels expands over time; a realistic but not widely-studied setting. We propose a deep learning technique for open-set learning based on **reserve output units (ROUs)**, which are designed to help a network anticipate the introduction of new categories during training. ROUs are additional output units whose representations are trained along with units for already-seen classes, which can be assigned to new classes once a labeled instance of a novel class is observed. We experiment with different initialization methods, compare this method with simply adding an new output vector for the novel class, and find that ROUs achieve better and more consistent performance than the simple add-new baseline. We further experiment with a loss which encourages the output space to match pretrained word embeddings, with the goal of encouraging good semantics of this space. In experiments on MNIST and CIFAR, this technique hurt or does not affect performance, but we're optimistic this technique could be helpful for larger output spaces.

## 1 RESERVE OUTPUT UNITS (ROUs)

A key question of open-set learning using deep networks is how to deal with the introduction of new categories. The most straightforward approach to incorporating a new category would be to use a standard classification architecture, and add a new output weight vector (and bias) for the new category, keeping all other parameters fixed. We call this **Add New** in the experiments section.

This approach, however, does not *anticipate* the introduction of novel categories, which our technique aims to do. Our method maintains a set of **Reserve Output Units (ROUs)** which do not correspond to any of the seen categories, but which the model *can* choose to predict (Figure 1).

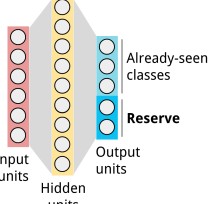 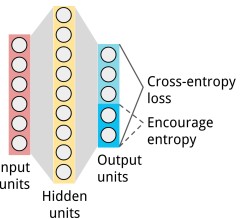 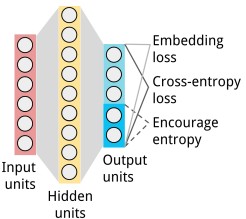

Figure 1: Reserve output units allow the network to anticipate the introduction of novel classes (**left**), because ROU representations are trained along with the those for already-seen classes. We use a standard categorical cross-entropy loss over all output units (including ROUs), and additionally penalize the entropy of softmax over just the ROUs, in order to encourage diverse representations (**middle**). We also experiment with a loss which encourages the final-layer hidden representations to be close to pretrained word embeddings, in order to encourage meaningful semantics (**right**). For architectural and training details see Section 4: Experiments.

When a new category is seen, we assign it to the ROU which assigns it the highest likelihood. We then *replace* the output vector of that ROU with either a random vector, or a scaled copy of the final hidden layer's activations on that new instance. We experiment with different methods of initializing and training ROUs, and find that using ROUs gives a consistent improvement over simply adding a new output vector.

## 2   OPEN-SET PROBLEM FORMULATION

In open-set learning, the set of labels or categories to learn expands over time. Formally, we define open-set learning formally as a classification problem with labels in $\mathbb{N}$. The algorithm is expected to correctly predict labels for **known** categories, and to correctly predict when an example is **novel**. There are thus three types of possible error, which the task loss may weight differently:

1. **Incorrect classification**: incorrectly classifying an example of a known class as a different known class
2. **False Positive (FP)**: incorrectly classifying an example of a known class as novel
3. **False Negative (FN)**: incorrectly classifying a novel example as an example of a known class

We consider that there are three important components of open-set learning:

1. Recognize that an example belongs to a novel class
2. Learn about and generalize to other examples of the novel class
3. Deal with unbalanced data

The problem can thus be viewed as anomaly detection followed by generalization to novel examples. The generalization step of open-set learning can be formulated either as one- or few-shot learning, or as an online learning problem. We follow the latter approach: at every step, the set of possible labels may (e.g. with some probability) be expanded to include a new category, and an example with one of the current possible labels is selected and presented to the learner.

## 3   BACKGROUND AND RELATED WORK

Scheirer et al. (2013) provide a review of open-set learning, and formulate a 1 vs. all Support Vector Machine for performing anomaly detection. (Bendale & Boult, 2015) perform the anomaly detection stage of open-set learning by using a modified softmax they call OpenMax, which adds an "unknown" category to a typical softmax layer. To our knowledge, our work is the first to address the problem of generalizing to novel classes with deep networks, and the first to view open-set learning as an online learning problem.

## 4   EXPERIMENTS

### 4.1   RESERVE OUTPUT UNITS VS ADD NEW

As a first experiment, we evaluate our method on MNIST, training a ReLU-MLP with 1 hidden layer of 100 units and no output biases. For ease of implementation, we fix the number of outputs (including reserve output units) at 20. At each step, a new example is seen and added to the learner's memory. We then train to convergence (or for 20 epochs, whichever is shorter) on a dataset which includes this observation and a set of 127 observations from memory, randomly sampled with replacement. We measure the total loss incurred over 1000 steps of learning, starting with one category and adding a new one every 100 steps.

### 4.2   EMBEDDING LOSS

In order to encourage meaningful semantics for the output space, we experiment with a loss that penalizes the distance between the activations of the final hidden layer and pretrained word embeddings for the target class. Specifically, we measure the cosine distance between a 300-D projection

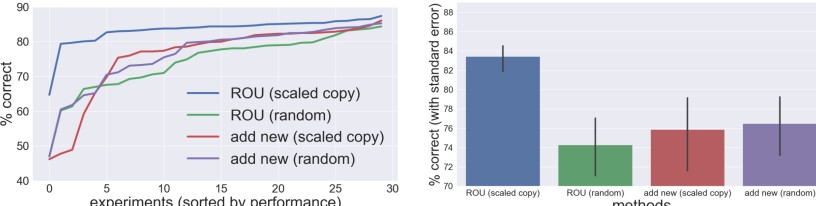

Figure 2: Comparison of **ROUs** to adding a new output unit each time a novel class is encountered (**add new**). Experiments(**Left**) plot shows that ROUs achieve better performance than Add New. Also, these experiments demonstrate scaling is important for both methods. Accuracy averaged over 30 training runs, with std error bars, (**right**) shows that ROUs also achieve much more consistent performance across experiments.

of the last layer's state and 300-D GloVe pretrained embeddings, and multiply this by a coefficient to weight its contribution to the loss. First, as a baseline, we experiment with this technique on MNIST, and find it degrades performance. This is unsurprising, since the word embeddings for different numbers are presumably very similar, and forcing the output vectors to be similar will hurt the ability of the model to discriminate between classes. We then perform experiments on CIFAR10, where the classes are more semantically distinct. We follow the procedure for the MNIST experiments, comparing with and without the embedding loss (with coefficient 1). Here we find that the embedding loss neither helps nor hurts; both methods achieve a performance of approximately 30% accuracy.

## 5 CONCLUSION

### 5.1 CONTRIBUTIONS OF THIS WORK

We formulate open set learning as an online learning problem, and propose reserve output units (ROUs) as a novel way for deep networks to anticipate the introduction of new classes over the course of training. We find that networks with ROUs achieve better, more consistent performance than a naive baseline of adding a new output unit each time a new class is encountered. We experiment with methods for initializing and training ROU representations, and find appropriate scaling of the representations to be very important. We performed initial experiments with a method of encouraging output representations to match pretrained word embeddings, but found this was not beneficial for MNIST or CIFAR10.

### 5.2 FUTURE WORK

We plan to test ROUs on a larger range of tasks which can be formulated as online learning problems, including sequential data. We are optimistic that the embedding loss method will be helpful for classification problems with larger output spaces, where having good semantics might be more important for learning. Specifically, we plan to test a variety of distance methods on an online learning version of ImageNet.

A more ambitious goal for this line of work is to leverage word-embeddings to perform zero-shot generalization to unseen classes, based on the similarity of embeddings in the hidden states of the image classifier. For instance, we might imagine that a network trained on images of "king", "man", and "woman" (and other categories) might be able to make a correct guess about the first "queen" it sees.

### ACKNOWLEDGMENTS

The authors are grateful to the CVPR Open Set workshop organizers for inspiration, and to the developers of Theano, Lasagne, and Keras. Tegan would like to thank NSERC (National Sciences Research Council of Canada) for funding that made this work possible.

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
