# OpenReview forum: "Reserve Output Units for Deep Open-Set Learning"
_ICLR.cc/2018/Workshop — Reject_

### Official Review · AnonReviewer3 · 2018-03-09
**Missing quite some significant comparison with previous works.**

**Rating:** 4
**Confidence:** 5

**Review:**

this paper tried to solve open-set learning by using the deep networks. It is claimed to be " the first to address the problem of generalizing to novel classes with deep networks, and the first to view open-set learning as an online learning problem."

However, this is definitely not true. The whole idea has been well studied more years before. Please check:
[1] T. Mensink, J. Verbeek, F. Perronnin, and G. Csurka. Distance-based image classification: Generalizing to new classes at near zero cost. TPAMI, 2013
[2] iCaRL: Incremental Classifier and Representation Learning, CVPR 2018

Also it would be great to compare with W-SVM and OPenmax.

Since the workshop is more focused on brilliant ideas, I donot think the idea of this paper if the first one proposed.

---

### Official Review · AnonReviewer2 · 2018-03-13
**The paper is very unclear**

**Rating:** 5
**Confidence:** 2

**Review:**

I'm unable to understand how the ROUs are participating to accounts for new words at test time. Figure 1 presents terms such as 'encourage entropy' but they don't feature in the text.

In this paragraph
"When a new category is seen, we assign it to the ROU which assigns it the highest likelihood. We
then replace the output vector of that ROU with either a random vector, or a scaled copy of the final
hidden layer’s activations on that new instance. We experiment with different methods of initializing
and training ROUs, and find that using ROUs gives a consistent improvement over simply adding a
new output vector.", what do the authors mean by output vector ? What is the motivation behind using random vectors or scaled copies of the final hidden layer's activations.

I'm also unclear on how the model is trained. Is it that during training, they introduce new words every couple of steps ? The results show that their approaches outperform the baseline, but it's not clear on what dataset and what task. Is this language modeling ? If yes, what were your perplexities.

There might be some very interesting ideas in the paper, but it needs an earnest rewrite to help us understand what is going on.

---

### Decision · Program_Chairs · 2018-03-20
**ICLR 2018 Workshop Acceptance Decision**

**Decision:**

Reject

**Comment:**

Based on the reviews, this paper has not been accepted for presentation at the ICLR workshop. However, the conversation and updates can continue to appear here on OpenReview.